# The Role of Uric Acid in Acute and Chronic Coronary Syndromes

**DOI:** 10.3390/jcm10204750

**Published:** 2021-10-16

**Authors:** Alessandro Maloberti, Marco Biolcati, Giacomo Ruzzenenti, Valentina Giani, Filippo Leidi, Massimiliano Monticelli, Michela Algeri, Sara Scarpellini, Stefano Nava, Francesco Soriano, Jacopo Oreglia, Alice Sacco, Nuccia Morici, Fabrizio Oliva, Federica Piani, Claudio Borghi, Cristina Giannattasio

**Affiliations:** 1School of Medicine and Surgery, University of Milano-Bicocca, 20126 Milan, Italy; marco.biolcati@ospedaleniguarda.it (M.B.); giacomo.ruzzenenti@ospedaleniguarda.it (G.R.); valentina.giani@ospedaleniguarda.it (V.G.); Filippo.leidi@ospedaleniguarda.it (F.L.); massimiliano.monticelli@ospedaleniguarda.it (M.M.); cristina.giannattasio@ospedaleniguarda.it (C.G.); 2Cardiology 4, ASST GOM Niguarda Hospital, 20121 Milan, Italy; michela.algeri@ospedaleniguarda.it (M.A.); sara.scarpellini@ospedaleniguarda.it (S.S.); 3Cardiology 1, ASST GOM Niguarda Hospital, 20121 Milan, Italy; stefano.nava@ospedaleniguarda.it (S.N.); francesco.soriano@ospedaleniguarda.it (F.S.); jacopo.oreglia@ospedaleniguarda.it (J.O.); alice.sacco@ospedaleniguarda.it (A.S.); nuccia.morici@ospedaleniguarda.it (N.M.); fabrizio.oliva@ospedaleniguarda.it (F.O.); 4School of Medicine and Surgery, University of Bologna—IRCCS Policlinico S. Orsola, 40138 Bologna, Italy; federica.piani2@unibo.it (F.P.); claudio.borghi@unibo.it (C.B.)

**Keywords:** uric acid, acute coronary syndrome, chronic coronary syndrome

## Abstract

Uric acid (UA) is the final product of the catabolism of endogenous and exogenous purine nucleotides. While its association with articular gout and kidney disease has been known for a long time, new data have demonstrated that UA is also related to cardiovascular (CV) diseases. UA has been identified as a significant determinant of many different outcomes, such as all-cause and CV mortality, and also of CV events (mainly Acute Coronary Syndromes (ACS) and even strokes). Furthermore, UA has been related to the development of Heart Failure, and to a higher mortality in decompensated patients, as well as to the onset of atrial fibrillation. After a brief introduction on the general role of UA in CV disorders, this review will be focused on UA’s relationship with CV outcomes, as well as on the specific features of patients with ACS and Chronic Coronary Syndrome. Finally, two issues which remain open will be discussed: the first is about the identification of a CV UA cut-off value, while the second concerns the possibility that the pharmacological reduction of UA is able to lower the incidence of CV events.

## 1. Introduction

Uric Acid (UA) is the final product of the catabolism of purine nucleotides from endogenous (cellular nucleoproteins) and exogenous origins (alimentary). Its biosynthesis, which principally involves the liver, also includes the gut, muscles and kidneys; urinary excretion is the main mechanism of UA elimination, while a small percentage thereof is removed by the intestine. At a pH of 7.4, the solubility limit of plasma UA is 6.8 mg/dL. Beyond this level, the conditions for urate crystal precipitation are created. Conditions which may raise UA levels include the increased production which occurs with a purine-rich diet, tumor lysis syndrome or specific drugs (chemotherapy and pyrazinamide), and also with a decrease in UA excretion, mainly in renal diseases. Genetics could also be a cause for hyperuricemia, as it happens in the gain of function of the enzyme phosphoribosyl-pyrophosphate synthetase, or in the deficit of hypoxantine-guanine phosphoribosyltransferase (completely in the Lesch-Nyhan syndrome, partially in the Kelley-Seegmiller syndrome) [1].

The association of UA with articular gout and kidney disease has been known for a long time, while new data has demonstrated that UA is also related to cardiovascular (CV) diseases. [1] In fact, UA was identified as a significant determinant of many different outcomes in the CV area, such as all-cause and CV mortality [2], and also of CV events (mainly Acute Coronary Syndrome—ACS) and stroke [3,4]. Furthermore, UA is correlated with the development of Heart Failure (HF) [5], and with a higher mortality in this group of patients [6], as well as with the onset of atrial fibrillation [7]. All of these significant findings led the latest European Guidelines of Arterial Hypertension to introduce UA among CV risk factors that should be assessed in order to stratify a patient’s risk [8].

Hyperuricemia represents an epidemiological problem, especially if CV comorbidities are present. Its prevalence ranges from 6% in healthy subjects [9] to 14% in hypertensives [10], with a significant increase to 23% among patients with ACS and Chronic Coronary Syndrome (CCS) [11,12]. After a brief introduction on UA’s general role in CV events, this review focuses on UA’s relationship with CV outcomes, as well as on specific features of patients with ACS and CCS. Despite the important number of publications on this topic, two issues remain open: the first is about the identification of a CV UA cut-off value, while the second concerns the possibility that the pharmacological reduction of UA is able to lower the incidence of CV events. These two fundamental points will also be discussed in this research paper.

## 2. Uric Acid and Cardiovascular Events

The connection between UA and CV events was demonstrated for the first time in 1967 by Kannel et al. [13] in the Framingham study, which included 5127 subjects with a 12-year follow-up, in which an increased risk of Myocardial Infarction (MI) was identified in subjects with hyperuricemia. Since this pivotal work, many other later publications confirmed this association, and hyperuricemia was recognized as an independent CV risk factor, and also when added to the traditional ones [1]. In fact, one of the biggest meta-analyses on this issue, including 29 prospective studies (for a total of 958,410 individuals), found a Hazard Ratio (HR) of 1.13 (95% CI 1.05–1.21) for MI, and of 1.27 (95% CI 1.16–1.39) for CV mortality in hyperuricemic patients [14].

The mechanisms by which UA could determine CV events have not been definitively identified. However, UA certainly acts at multiple levels, as shown in Figure 1.

First, the oxidative stress determined by the two final biochemical reactions is involved: during the conversion of hypoxanthine into xanthine (and hence into UA) determined by the xanthine oxidase enzyme, the generated superoxide anions increase oxidative stress, a well-known atherosclerotic risk factor [15,16]. In addition, concurrent processes induced by xanthine oxidase include the oxidative role of NADH and the nitrate reduction activity [17], which are two other factors which are able to induce oxidative stress. This leads to the “xanthine oxidase theory”, according to which the decrease of UA through the inhibition of the enzyme, instead of an increased renal elimination of UA, is most beneficial in terms of CV risk reduction.

Oxidative stress represents a fundamental pathway in diseases related to hyperuricemia (hypertension and Diabetes Mellitus—DM), and in the development of heart and vessels organ damage. In fact, UA has been linked to the development of arterial hypertension [18], DM [19], and metabolic syndrome [20], which in turn increase the rate of CV events. The molecular paths possibly explaining the relationship between UA and hypertension include the activation of the renin-angiotensin-aldosterone system [21] and an impairment in endothelial function due to a reduction of nitric oxide levels [22]. Regarding metabolic derangement, UA is involved in the deamination of adenosine monophosphate, resulting in increased fat accumulation, which is one of the steps at the basis of hyperinsulinemia, and consequently of insulin resistance [23]. In addition, UA can block oxide nitric-mediated insulin release, and increases the oxidative damage in pancreatic B-cells [24].

One of the most important mechanisms through which UA is probably related to CV events is renal damage, a well-known CV risk factor. UA can affect kidneys by depositing crystals in renal tubules during hyperuricosuria [25], resulting, together with increased oxidative stress, in tubule-interstitial inflammation with afferent arteriopathy of the arteriole and hyperplasia/hypertrophy of the tunica muscularis [26]. In fact, UA is definitely related to the reduction of the glomerular filtration rate, as well as to microalbuminuria [27]. However, the link between UA and kidney damage is certainly a two-way correlation, as the loss of renal function results in the decreased excretion capacity of UA with increased plasma levels [28].

Finally, UA also appears to be linked to heart and vessel damages. In particular, the relationship with pulse wave velocity, which is the most widely used measurement of arterial stiffness, suggests a possible association between UA and changes in vascular structure and function [29].

Another relevant aspect in the relation between UA and CV diseases is the role of gender. Some studies describe a link with target organ damage only in females [11], while CV outcomes (all-cause mortality, CV mortality and ACS) seem to be related to UA only in females in some studies [30] and only in males in others [31]. Some possible explanations include the existence of gender differences in the gene functions controlling the biochemical pathways of UA [32], but also the role of hormones in women and their involvement in UA metabolism. In fact, hyperuricemia has been associated with a higher left ventricular mass index during post-menopause, but not in pre-menopause [33].

## 3. Uric Acid and Acute Coronary Syndrome

As was already mentioned, hyperuricemia appears to be associated with fatal and non-fatal ACS in the general population [4]. In patients experiencing an ACS, it is a common finding (reported in 23% of the total subjects [11]). Moreover, in patients admitted for an ACS, UA seems to be related to in-hospital [11] and long term [34] all-cause and CV mortality, and also to higher rates of in-hospital adverse events (such as atrial fibrillation or bleeding [35]) and to longer inpatient stays.

In particular, many studies found an association between UA levels at admission and specific HF-related issues, such as the Killip class, the use of intra-aortic balloon pump and cardiogenic shock, and a reduced left ventricular ejection fraction at admission [11,36]. This last point raises an interesting question, i.e., whether UA in ACS is a determinant of worse presentation or simply a marker of a poorer condition. In other words, is UA a significant determinant or just an innocent bystander in the context of an ACS? To date, the answer is not totally clear, and both supporting and non-supporting data has been published (Table 1). Furthermore, most knowledge on this issue is derived from cross-sectional and prospective studies, while only a few Randomized Clinical Trials have been published (and will be discussed further on).

Some studies identify UA as a determinant of a more severe coronary artery involvement, a larger infarct size [37], a greater risk of acute plaque complications (such as the formation of a completely obstructing thrombus) [38] and a higher prevalence of challenging revascularization procedures, which could remain incomplete [39]. It is also possible that the UA increase in HF is merely an epiphenomenon of the cardiac damage, and is not the triggering cause, as it could secondarily rise due to increased purine metabolism caused by hypoxia and tissue catabolism [40], enhanced purine release from ischemic cells (both from the heart and from peripheral hypoperfused tissue) and the reduced clearance deriving from ACS-related impaired renal function. Furthermore, a hyperactivation in xanthine oxidase activity was also found in acute decompensated HF [41]. Finally, patients with HF-related issues during ACS make use of diuretics more frequently; the latter are a well-known iatrogenic cause of hyperuricemia [42].

If the assumption that UA acutely increases during hospitalization due to secondary hemodynamic effects is true, a decrease in its values from admission to discharge should be expected. However, data on longitudinal UA changes in ACS subjects are still lacking.

## 4. Uric Acid and Chronic Coronary Syndrome

Hyperuricemia is a significant epidemiological problem in CCS, and has been strongly connected to CV mortality and CV events in this specific subgroup of patients [43,44,45,46,47]. The most important issue in these subjects is the possible relationship between UA levels and the extent and severity of Coronary Artery Disease (CAD). As was also shown in Table 2, most of the publications on this topic show that UA correlates with CAD, as defined as both the number of vessels involved [48] and specific scores, such as Gensan [49] or Syntax [50]. However, other studies did not identify this association [9,51,52]. This heterogeneity may be explained by differences in the sample selection (never revascularized, newly diagnosed patients versus individuals with previous MI and/or previous coronary revascularization) and in the assessment of the CAD. In this realm, studies that assessed CAD only in terms of the number of damaged vessels, without taking into account more sensitive scores [51,52], did not find any significant correlation. Furthermore, other surveys which considered newly diagnosed and never treated subjects more frequently found a positive association [53,54]; by contrast, when patients presented a positive anamnesis for previous MI/revascularization [9] or a strong risk factor (such as DM [55]), the association lacked. Taken together, these findings lead to the hypothesis that UA could act on coronary arteries only in an early phase of the atherosclerosis disease, through the various mechanisms seen in Section 2. In other words, when CAD progresses to a more advanced stage, other factors (such as previous MI, previous myocardial revascularization, DM) may overshadow the effects of UA and limit the possibility of finding a significant association with CAD. Thus, in a group of patients with very high CV risk, the presence/absence of hyperuricemia may not change further the overall risk profile.

Gender could be another factor influencing the UA–CAD association. Only two studies carried out a separate analysis in males and females, finding a connection only in the latter [56]. In addition, a piece of research based on non-menopausal females only confirmed the association with the severity of CAD [57].

**Table 1 jcm-10-04750-t001:** Summary of the available data on the association between uric acid and Acute Coronary Syndrome.

Study	Type of Study	Supporting Data	Non-Supporting Data	Reference
Bos et al.	Prospective cohort study	Significant association between baseline UA and risk of both CAD and stroke, only slightly attenuated by adjustment for other CV risk factors		[4]
Centola et al.	Prospective cohort study	High admission levels of UA are independently associated with in-hospital adverse outcomes and mortality of ACS patients		[11]
Mehmet ed al.	Prospective cohort study	Elevated UA levels on admission are independently associated with impaired coronary flow after primary PCI and both short-term and long-term outcomes in patients who undergo primary PCI for the management of STEMI		[34]
Nadkar et al.	Case control study	UA levels are higher in patients with acute MI and correlate with Killip class.		[36]
Kobayashi et al.	Prospective cohort study	High UA levels are the primary predictor of 2-year cardiac mortality.		[37]
Lazzeri et al.	Prospective cohort study	UA levels are associated to greater risk of acute plaque complications		[38]
Okazaki et al.	Case control study		Plasma XOR activity was extremely high in patients with severely decompensated AHF, in association with a high lactate value and leading eventually to hyperuricaemia	[40]
Maloberti et al.	Prospective cohort study		Diuretic-related hyperuricemia carry a similar risk of CV events and all-cause mortality when compared with individuals that present hyperuricemia in absence of diuretic therapy	[42]

XOR: Xanthine Oxido-Reductase. AHF: Acute Heart Failure. CAD: Coronary Artery Disease. CV: cardiovascular. UA: Uric acid. ACS: Acute Coronary Syndromes. PCI: Percutaneous Coronary Intervention. STEMI: ST-Elevated Myocardial Infarction.

**Table 2 jcm-10-04750-t002:** Summary of the available data on the association between uric acid and Chronic Coronary Syndrome.

Study	Type of Study	Supporting Data	Non-Supporting Data	Reference
Okura et al.	Population based cohort study	Elevated UA is an independent predictor of CV events and all-cause mortality combined in patients with CCS		[43]
Tian et al.	Population based cohort study	UA levels were associated with the presence and severity of CAD; UA may be involved in the progression of CCS.		[48]
Duran et al.	Population based cohort study	UA was significantly associated with number of diseased vessels and is an independent risk factor for multivessel disease.		[49]
Karabağ et al.	Population based cohort study	UA was to be associated with high Syntax Score and long-term mortality in patients with MVD		[50]
Tasić et al.	Population based cohort study		Asymptomatic hyperuricemia is not significantly associated with the severity of CAD	[51]
Zand et al.	Case control study		UA is not an independent risk factor for premature CAD but is weakly correlated with the extent of the disease	[52]
Verdoia et al.	Population based cohort study		Among diabetic patients, higher UA is not independently associated with the extent of CAD or with platelet aggregation.	[55]
Maloberti et al.	Population based cohort study		UA do not play a role in determining coronary arteries disease as well as LV diastolic dysfunction in CCS subjects	[12]

CCS: Chronic Coronary Syndrome. MVD: Multi Vessel Disease.

## 5. The First Open Question: The Cardiovascular Cut-Off

The commonly-used cut-offs of 6 mg/dL in women and 7 mg/dL in men were established on evidence regarding gouty patients, rather than CV events in asymptomatic hyperuricemia. These thresholds are based on the UA saturation point (6.8 mg/dl at a pH of 7.4) which determines its precipitation in joints and kidneys, leading to the classic form of gout. However, previous evidence suggests that UA could act negatively on the CV system even at lower serum levels [1], as crystal precipitation is just one of the possible causes determining the relationship between UA and CV events.

Despite the large number of published studies, the identification of a CV UA cut-off value is still a matter of discussion. Among others, recently published results from an Italian multicenter, retrospective, observational cohort study brought new light on the CV cut-off question. The URRAH project (Uric acid Right for heArt Health) entailed data collection on outpatients (mainly hypertensives) and the general population, with a total of 23,475 subjects and a follow-up period of 20 years. Regarding all-cause mortality, a threshold of 4.7 mg/dL was detected, while 5.6 mg/dL emerged as the most suitable cut-off for CV mortality [58]. In both cases, the addition of the UA to the CV risk scores determined a significant increase of the area under the curve, leading to a re-classification of 33% of outpatients and 40% of the general population subjects. Concerning the specific analysis on MI, similar thresholds emerged: according to gender, cut-offs of 5.27 mg/dL in women and 5.49 mg/dL in men were identified [59]. In addition, a further threshold of 4.89 mg/dL was found to be predictive of fatal HF in another specific analysis [5]. These data come from population studies, while, despite our specific focus, no paper has ever been published about CCS. In acute events, subjects with UA > 7.5 mg/dL reported a significant subsequent mortality, but with low sensibility and specificity (0.64 and 0.66 respectively) [60]. In spite of the differences according to the considered outcome, all of these values emerged as being much lower than the conventional hyperuricemia cut-offs (Figure 2). That a lower cut-off should be used when evaluating the relationship between UA and CV outcomes is, to date, undoubted. However, what is already known is insufficient to recommend a single UA threshold for CV risk. While waiting for further results on this issue, a more suitable cut-off must be chosen on an individual basis strongly, depending on the patient’s CV risk and previous CV events.

Another interesting point is the identification, in some studies, of a J-curve in the relationship between UA levels and CV events, meaning not only that high levels of UA raise the risk but also that too-low values could be harmful [61,62]. For example, according to the largest of these works (which included 127,771 subjects), an increased risk was detected in individuals with hyperuricemia, and also in subjects with circulating UA levels below 4 mg/dL. This occurrence showed differences in terms of its statistical correlation depending on gender: while women showed a linear trend in the link between UA and all-cause mortality, a J-shaped association was found in men, in which a lower-cut off of 3.4 mg/dL was identified as a significant threshold of adverse outcomes [62]. This evidence can be possibly interpreted as being due to the fact that, besides the potential pro-oxidative role, UA also has anti-oxidant properties which contribute to scavenging reactive oxygen species, chelating transition metals, and preventing the degradation of superoxide dismutase [63].

## 6. The Second Open Question: Are Uric-Acid-Lowering Therapies Effective in Reducing the Risk of Cardiovascular Events?

Currently, the therapies for the reduction of UA levels are exclusively recommended in patients with hyperuricemia associated with gouty arthritis or gouty nephropathy; these include xanthine oxidase inhibitors (Allopurinol and Febuxostat) and uricosurics (Probenecid and Lenisurad). Xanthine oxidase inhibitors were the first registered drugs, and Febuxostat showed the greatest inhibiting effectiveness with a complete selectivity for xanthine oxidase, while allopurinol also works on other enzymes involved in purine metabolism. Instead, Probenecid and Lenisurad increase the urinary excretion of UA acting on a specific renal transporter. Many other molecules are under development [64], but they still require further studies before becoming available. The reduction of UA achieved by these drugs has been demonstrated to reduce the amount of gout exacerbation and the disease severity [65], but whether it leads also to a decrease in CV morbidity and mortality is still a matter of debate. Currently, few large and randomized studies using CV events as a primary outcome have been published, and most of the evidence is based on pre-clinical investigations or studies on humans with surrogate end-points. For example, allopurinol was found to be able to lower blood pressure [66] and reduce subclinical organ damage (in particular intima-media thickness [67] and left ventricular mass index [68]), and this could theoretically lead to a possible reduction of CV events. Furthermore, experimental evidence suggests that allopurinol improves mechano-energetic uncoupling in the myocardium, thus decreasing myocardial oxygen consumption [69], and might be beneficial to patients with cardiac ischemia and angina. Possible explanations include the prevention of oxygen wastage for the avoidance of its consumption, due to the inhibition of xanthine-oxidase, and an improvement in microvascular function thanks to its positive effects on endothelial function [70]. However, there are currently no data regarding the possible link between these effects and the reduction in UA determined by allopurinol [71].

Some observational studies reported a decrease of CV events in patients treated with hypouricemic drugs [72], but, as is well known, this kind of design implies a high probability of bias; thus, these preliminary results need to be confirmed by double-blinded Randomized Clinical Trials (RCT) in order to exclude the presence of confounding factors (Table 3).

Two RCTs limited to individuals with gout have been published, showing opposite results. The CARES (Cardiovascular safety of Febuxostat and Allopurinol in patients with gout and cardiovascular comorbidities) study randomized 6190 patients with gout and previous CV events to Allopurinol vs. Febuxostat [73]. The subjects treated with Febuxostat reported a greater number of CV events, which worried the scientific community and consequently induced national agencies for drugs safety worldwide to warn about its use in patients with prior MI. In 2018, the FAST (Febuxostat versus Allopurinol Streamlined Trial) study, performed on 6128 gouty patients without prior CV events randomized to Allopurinol versus Febuxostat, found the non-inferiority of the latter with respect to the primary CV endpoint (a composite of hospitalization for non-fatal CV events and CV death) [74]. An extensive discussion about the differences between the two RCTs is beyond the purpose of this review, as a focused paper has been published on this topic [75]. However, the CARES trial presents many important biases; foremost, and also valid for the FAST trial, is that the absence of a control group disallows us to conclude whether Allopurinol reduces CV risk or Febuxostat raises it. Secondly, more than 50% of the individuals discontinued the therapy within the first year from enrolment, and although this trend was comparable between the two groups, no information about the following administered drugs was provided. As the patients were symptomatic for gout, it is likely that another drug had been prescribed (e.g., shifting Allopurinol to Febuxostat or vice-versa, or introducing a uricosuric). Similarly, no data were supplied on specific therapies about the other CV risk factors (hypertension, DM and dyslipidemia) which could underlie the differences in the CV events. In conclusion, these studies are not sufficient to demonstrate the superiority of one approach over the other in patients with gout, and, furthermore, they did not take into consideration asymptomatic patients with hyperuricemia.

For this latter group, one study is on-going and another one has been already published. The FREED (Febuxostat for cerebral and caRdiorenovascular events PrEvEntion StuDy) trial [76] compared Febuxostat with other treatments in 1,070 subjects without gout but with high CV risk or previous CV events. The main findings include the absence of differences in CV events and mortality, but with a significant reduction in renal events (defined as new-onset microalbuminuria or its progression) in the Febuxostat-treated patients. The on-going ALL-HEART (ALLopurinol and Cardiovascular Outcomes in Patients with Ischemic HEART Disease) study [77] already randomized 5,938 patients with CCS to Allopurinol versus placebo; its results are particularly awaited because they will determine whether Allopurinol improves major CV outcomes in patients with CCS, thereby changing the paradigm of secondary CV prevention strategies. Until it is concluded, the treatment of hyperuricemic individuals without gout is clinically not recommended.

Unfortunately, no information about the efficacy of UA reduction in primary CV prevention is available because of the lack of studies on asymptomatic hyperuricemia in individuals without prior CV events.

Eventually, regarding ACS, one specific but small and non-randomized study is available [78]; it enrolled 50 patients that started allopurinol for clinical indications within 14 days from ACS admission, and another 50 individuals that were not in therapy as a control group. During the 2-year follow-up period, inflammatory biomarkers were significantly lowered in the allopurinol group, as well as the number of CV events (10% vs. 30% for allopurinol and the control group, respectively). However, a larger sample is needed in the subset of ACS patients to confirm this possible benefit.

**Table 3 jcm-10-04750-t003:** Studies on the relationship between drugs acting on UA and their effects on CV diseases.

Study	No of Participants	Drugs Compared	Outcomes	Results	Reference
CARES	6198	Febuxostat vs. Allopurinol	4-component MACE (CV death, non-fatal MI, nonfatal stroke and unstable angina with urgent coronary revascularization)	Febuxostat is associated to a greater number of CV events	[73]
FAST	6128	Febuxostat vs. Allopurinol	Composite of hospitalization for non-fatal MI or biomarker-positive ACS; non-fatal stroke; CV death	Febuxostat is non-inferior to allopurinol	[74]
FREED	1070	Febuxostat vs. Other treatments	Composite of cerebral or cardiorenovascular events, all deaths	Febuxostat is associated to a redu-ction in renal events.	[76]
ALL-HEART	5938	Allopurinol vs. placebo	Composite of non-fatal MI, non-fatal stroke or CV death	On going	[77]
Huang et al.	100	Allopurinol vs. placebo	CV events	Allopurinol reduces inflammatory biomarkers and CV events	[78]

MACE: Major Adverse Cardiovascular Events.

## 7. Conclusions

Although many findings have been published in favor of a role of UA in CV diseases (particularly in ACS and CCS), several points remain not completely understood in this complicated relationship. From a pathophysiological point of view, the question of whether hyperuricemia contributes directly to the genesis of ACS and CCS, or if it is just an innocent bystander determined by an increased catabolism in ischemic myocardium is still a matter of debate. Furthermore, the lack of an unequivocally accepted UA CV cut-off does not allow us to define a clear threshold of CV events’ risk and their fatality. While waiting for further results on this issue, a more suitable cut-off must be chosen on an individual basis, strongly depending on the patients’ CV risk and previous CV events. Finally, there is currently no strong evidence about a certain benefit deriving from a pharmacological treatment of hyperuricemia in terms of the reduction of CV morbidity and mortality. The on-going ALL-HEART trial will provide us with an important answer, by determining whether allopurinol reduces CV events in patients with CCS, and it could perhaps change the role of UA in secondary CV prevention strategies. In conclusion, more studies should be performed in order to clarify the involvement of this molecule in the spectrum of CV disease and its possible role as a target in CV prevention strategies.

## Figures and Tables

**Figure 1 jcm-10-04750-f001:**
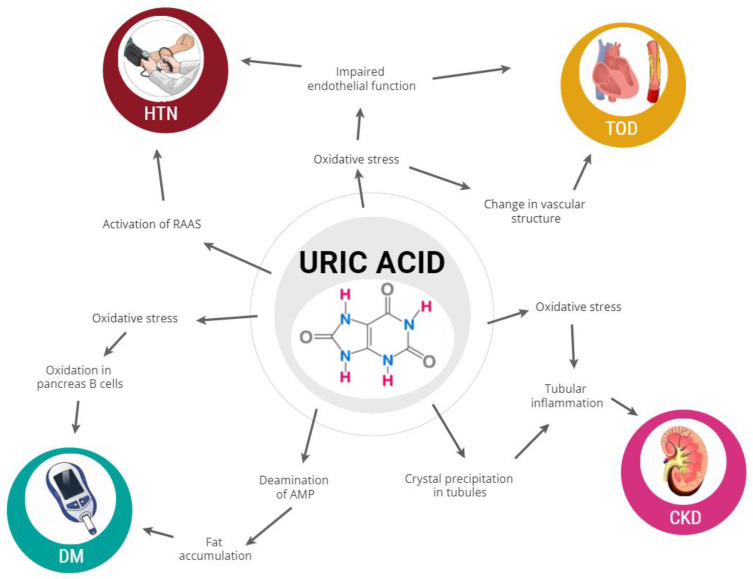
Mechanisms contributing to the relationship between uric acid and cardiovascular diseases. HTN = Arterial Hypertension; DM = Diabetes Mellitus; CKD = Chronic Kidney Disease; TOD = Target Organ Damage.

**Figure 2 jcm-10-04750-f002:**
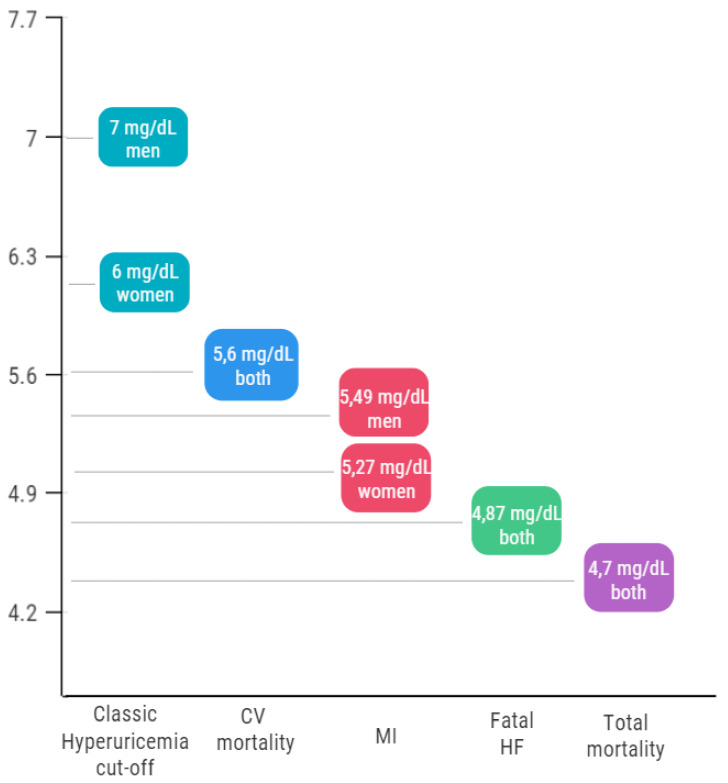
Summary of the different uric acid cut-offs according to cardiovascular diseases. CV = Cardiovascular; MI = Myocardial Infarction; HF = Heart Failure.

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
