# Peer review of "The Role of Uric Acid in Acute and Chronic Coronary Syndromes"

_jcm, 2021, doi:10.3390/jcm10204750_

Round 1

Reviewer 1 Report

I read with interest the review by Maloberti and colleagues.
It is focused on uric acid and its relationship with cardiovascular outcomes. The review is well written, the topic is actual.

I have few suggestions:

  1. Experimental evidence suggests that allopurinol lowers myocardial oxygen consumption and might be beneficial in patients with cardiac ischemia and angina. What is the mechanism? Does this anti-anginal effect correlate with UA serum levels? Please consider to implement paragraph 4 or 6.
  2. A table which summarizes the studies reported in paragraph 6 could be useful.
  3. Typo at paragraph 3, line 5 ("higher rats of in-hospital...")

Author Response

Dear Editor,

we have to thanks the reviewer for the time and energy that invested in our manuscript. We believe that the comments and suggestions provided in the revision, gave us the opportunity to improve the manuscript considerably and better present the results of our study. Please find below the answers to all of the comments point by point. In the revised manuscript, the respective changes have been highlighted in yellow.

  • Experimental evidence suggests that allopurinol lowers myocardial oxygen consumption and might be beneficial in patients with cardiac ischemia and angina. What is the mechanism? Does this anti-anginal effect correlate with UA serum levels? Please consider to implement paragraph 4 or 6.
  • The reviewer raised an important issue that have been now added to the chapter number 6. Insight into mechanisms and its relationship with UA reduction, togheter with 3 ne references, have been added (page 9, line 273).

  • A table which summarizes the studies reported in paragraph 6 could be useful.
  • A new table (number 3) that summarized the results of studies reported in chapter 6 have been added. Furthrmore, following the suggestion of reviewer 2, similar tables also for the previous chapter have been added (table 1 and table 2).
  •  
  •  
  • Typo at paragraph 3, line 5 ("higher rats of in-hospital...")
  • Thank you for your careful reading. Typos have been corrected and, following also the suggestion of reviewer 2, the text have been fully revised for other errors and English grammar.

Reviewer 2 Report

In this narrative review, Dr. Maloberti and colleagues discussed the role of uric acid in both acute and chronic coronary syndromes, and exposed the readers to two difficult yet relevant questions to be answered in future research. Overall, this could be a nice and insightful review. However, several concerns need to be addressed by the authors:

  • I would encourage the authors to carefully check for grammatical errors and typos. There are too many errors in this manuscript that sometimes alter the readability and clarity of the content. An assistance from a native English writer or a professional editing service might be needed. 
  • "Cardio-Vascular" doesn't need hyphen in the middle. It should be "cardiovascular"
  • In general, the authors need to add a few sentences at the beginning of the introduction about the source of uric acid and which conditions could possibly increase the amount in the body up to the pathological level.
  • "Its association with articular gout and kidney disease has been known for a long time, while most recent data have found UA to be related also to Cardio-Vascular (CV) diseases.". Please add a citation to this statement.
  • "an epidemiological important problem"
  • "It ranges from 6% in healthy subjects [8] to 14% in hypertensives [9],", was this incidence? prevalence? or mortality? please clarify.
  • The explanation about mechanisms of UA-induced damage in Figure 1 and the following paragraphs are not clear and need to be reconstructed. The authors put oxidative stress at the same level as diseases, such as hypertension, DM etc. However, it could be that oxidative stress also plays a role in the generation of those diseases. So, the figure is not accurate.
  • Also, the explanation about UA and oxidative stress below figure 1 can be better elaborated. Please improve the clarity of this paragraph.
  • I would suggest the authors to describe the details of the mechanisms in a figure which also includes RAAS activation and its downstream effects. This is important to unravel the pathomechanisms of UA in inducing CV and kidney diseases.
  • "In other word, is UA a significant determinant or just an innocent bystander in the context of an ACS?" leaving a hanging question in a review is not advisable. This is actually the objective of this review to enlighten us with regards to this particular question, so even if there is no clear answer, the authors still need to add some insights and potential answers.
  • I believe that this topic about UA and CVD is interesting but also controversial. Therefore, I would expect that the authors could convince the readers why UA could be more than just a bystander in CVD. This could be done by unraveling the mechanisms and discuss the available clinical data thoroughly. I would suggest the authors to create a table for both CCS and ACS elaborating the currently available connections between UA and both coronary syndromes. The authors need to describe all the supporting (pro) and non-supporting (contra) data against this possible link between UA and coronary syndromes.
  • Please also add the type of studies in the abovementioned table because it would be difficult to believe that UA has an effect on CVD through observational studies which are typically influenced by many confounding elements and interactions between variables. Including RCTs into the discussion would be great.
  • Please add a sentence at the end of section 5 on the cut-off value that the authors believe we should use based on the reviewed data. Do the authors think that 6 and 7 mg/dL are still valid and relevant, or they need to be adjusted based on data that has been exemplified by the authors? If the authors believe that the cut-off value has to be evaluated case-by-case, please also mention those values in this summary sentence. 
  • Similarly, please create a table describing the available data on the efficacy of UA lowering medications on CCS and ACS to know how many of them are actually supporting this notion and how many objected the significance of UA drugs.
  • Please also focus on the CCS and ACS since this is the objective of this review. I see some discussions about UA and HF. Unless the HF is associated with coronary syndromes, the HF only data needs to be removed. OR, the authors could change the title to be broader, discussing not only CCS and ACS but CVD in general. However, this means that the authors need to expand the scope of the literature review to include other CVDs, including cardiac arrhythmias and vascular diseases as well. 

Author Response

Dear Editor, 
we have to thanks the reviewer for the time and energy that invested in our manuscript. We believe that the comments and suggestions provided in the revision, gave us the opportunity to improve the manuscript considerably and better present the results of our study. Please find below the answers to all of the comments point by point. In the revised manuscript, the respective changes have been highlighted in yellow.

  • I would encourage the authors to carefully check for grammatical errors and typos. "Cardio-Vascular" doesn't need hyphen in the middle. It should be "cardiovascular". "an epidemiological important problem"
  • Thank you for your careful reading. The text have been fully revised for other errors and English grammar.

  • The authors need to add a few sentences at the beginning of the introduction about the source of uric acid and which conditions could possibly increase the amount in the body up to the pathological level.
  • As suggested, few sentences on the source of uric acid and on factors determining its increase have been added to the introduction (page 1, line 27).

  • "Its association with articular gout and kidney disease has been known for a long time, while most recent data have found UA to be related also to Cardio-Vascular (CV) diseases.". Please add a citation to this statement.
  • A specific reference have been added.

  • "It ranges from 6% in healthy subjects [8] to 14% in hypertensives [9],", was this incidence? prevalence? or mortality? please clarify.
  • We were talking about hyperuricemia prevalence. This have been now clarified.

  • The explanation about mechanisms of UA-induced damage in Figure 1 and the following paragraphs are not clear and need to be reconstructed. The authors put oxidative stress at the same level as diseases, such as hypertension, DM etc. However, it could be that oxidative stress also plays a role in the generation of those diseases. So, the figure is not accurate.
  • Also, the explanation about UA and oxidative stress below figure 1 can be better elaborated. Please improve the clarity of this paragraph.
  • I would suggest the authors to describe the details of the mechanisms in a figure which also includes RAAS activation and its downstream effects. This is important to unravel the pathomechanisms of UA in inducing CV and kidney diseases.
  • Figure 1 have been completely redrawed in order to include the suggestion of the reviwer. Also the paragraph on oxidative stress have been rewritten (page 3, line 87).
  •  
  • "In other word, is UA a significant determinant or just an innocent bystander in the context of an ACS?" leaving a hanging question in a review is not advisable. This is actually the objective of this review to enlighten us with regards to this particular question, so even if there is no clear answer, the authors still need to add some insights and potential answers.
  • I believe that this topic about UA and CVD is interesting but also controversial. Therefore, I would expect that the authors could convince the readers why UA could be more than just a bystander in CVD. This could be done by unraveling the mechanisms and discuss the available clinical data thoroughly. I would suggest the authors to create a table for both CCS and ACS elaborating the currently available connections between UA and both coronary syndromes. The authors need to describe all the supporting (pro) and non-supporting (contra) data against this possible link between UA and coronary syndromes.
  • Please also add the type of studies in the abovementioned table because it would be difficult to believe that UA has an effect on CVD through observational studies which are typically influenced by many confounding elements and interactions between variables. Including RCTs into the discussion would be great.
  • The reviewer is absolutely right regarding the need not to be ambiguous. However, data in this issue are scanty and all comes from prospective and case control studies, the only RCTs available regards therapeutic option and are discussed in chapter 6. This explanation have been added to the paper (page 4, line 138) as well as the two proposed tables with pro and contrary data.

  • Please add a sentence at the end of section 5 on the cut-off value that the authors believe we should use based on the reviewed data. Do the authors think that 6 and 7 mg/dL are still valid and relevant, or they need to be adjusted based on data that has been exemplified by the authors? If the authors believe that the cut-off value has to be evaluated case-by-case, please also mention those values in this summary sentence.
  • A new figure (number 2) have been added in order to summarized actual and recent founded cut-offs. Furthermore, some sentences suggesting the use of the new and lower cut-off have been added to the cut-off (page 7, line 225) and to the conclusion section (11, line 346).

  • Similarly, please create a table describing the available data on the efficacy of UA lowering medications on CCS and ACS to know how many of them are actually supporting this notion and how many objected the significance of UA drugs.
  • A specific table (number 3) have been created summarizing the data coming from RCTs. For the moment also this data are not completely informative the draw definitive conclusions since the most important of them (all-HEART study) is already ongoing. This was already clearly stated in the chapter 6 (page 10, line 316) and a specific paragraph have been added to the conclusion(page 11, line 350).

  • Please also focus on the CCS and ACS since this is the objective of this review. I see some discussions about UA and HF. Unless the HF is associated with coronary syndromes, the HF only data needs to be removed. OR, the authors could change the title to be broader, discussing not only CCS and ACS but CVD in general. However, this means that the authors need to expand the scope of the literature review to include other CVDs, including cardiac arrhythmias and vascular diseases as well.
  • Following reviewer suggestion, HF only data have been removed trough the entire paper.

Round 2

Reviewer 2 Report

Thank you for addressing my previous comments. I do still see some errors that need to be corrected:

  • The use of English is not entirely good or clear, therefore significant changes should be made to improve the quality of the writing. For example "Mechanisms contributing in the relationship between uric acid and cardiovascular diseases". Please re-check the whole manuscript since these errors often impair the readability of the manuscript. A help from native English writer could be needed. 
  • In the abstract, since HF and CCS abbreviations are only used once, they are not needed. Please remove them. 
  • Please change the capital "C" in "Cardiovascular" to small letter.
  • The sentences about the pathomechanisms of UA in lines 76-79 are confusing and not clear. Please rephrase. Not sure what it meant.
  • The figure legend of Figure 1 is now on top of the figure. Please correct.
  • "Oxidative stress represent a fundamental pathway in diseases related to hyperuricemia (hypertension and Diabetes Mellitus – DM) and in the development of heart and vessels organ damage." This sentence needs to be moved somewhere else (perhaps around line 87). It doesn't have any connection to the following sentence. 
  • "Taken together, these findings lead to the hypothesis that UA could act on coronary arteries in a first phase of the disease through the various mechanisms seen in the first chapter." which phase is the first phase? Please clarify. Also, I think "first chapter" should be first section? However, the mechanisms are located in section 2 of this manuscript? Please clarify as well.
  • "Another work based only on a population of non-menopausal females confirmed the association with severity of CAD in this sample" What sample? Maybe "population"?
  • "MVD: MultiVessle Disease" should be "vessel"
  • "However, what is already known is not sufficient to recommend a single UA threshold for CV risk." not sufficient should be insufficient
  • "Summary of the different uric adic..." should be uric acid

Author Response

We have to thank again the reviewer for him/her careful reading of our works. We have addressed all the grammar and typos issue he/she report. Furthermore, the paper have been fully revised to improve the quality of reading. Also, highlighted unclear sentences have been modified accordingly to reviewer suggestion. The respective changes have been highlighted with the track changes in the revised version. We hope that the revised manuscript meets also the expectations of the reviewer.